# Positive Resources for Flourishing: The Effect of Courage, Self-Esteem, and Career Adaptability in Adolescence

Anna Parola [1,*] and Jenny Marcionetti [2]

1  Department of Humanities, University of Naples Federico II, 80133 Naples, Italy
2  Department of Education and Learning, University of Applied Sciences and Arts of Southern Switzerland, 6928 Manno, Switzerland
*  Correspondence: anna.parola@unina.it

**Abstract:** Flourishing is defined as an optimal state of functioning in which individuals pursue their goals and aspirations. Hence, flourishing seems to be a protective factor for career transitions in adolescence. This study aimed to analyze the predicting role of self-esteem, courage, and the four career adaptability dimensions, i.e., concern, control, curiosity, and confidence, on flourishing. The sample consisted of 221 Italian adolescents attending the last year of middle school. The preliminary analyses showed gender differences in courage and flourishing, reporting females higher scores than males on both variables. The SEM path model showed that courage, self-esteem, and confidence predict flourishing, and suggested that confidence partially mediates the relationship between courage, self-esteem, and flourishing. Findings have also permitted us to draw practical implications for interventions in adolescence.

**Keywords:** flourishing; courage; self-esteem; career adaptabilities; career confidence; gender; adolescence





## 1. Introduction

Making career choices is the most demanding developmental task for adolescents and young people because it establishes a fundamental basis for psychological well-being [1]. During adolescence, individuals begin to focus on educational and vocational goals, and by the end of middle school, they are called upon to make their first career decision. Several studies have underlined the centrality of this task and its contribution to positive youth development [2,3].

To deeply understand adolescents' career decision-making process, it is important to consider the specific social and economic environment in which they take place [4]. Nowadays, managing career transitions means tackling the current threats and challenges that characterize the society. There are four main challenges. Firstly, technological evolution and digitalization have induced changes in the labor market putting some occupational sectors at risk. The latest data from World Economic Forum [5] estimated that 85 million jobs will be transferred from humans to machines, while 97 million new jobs will emerge in which there will be a collaboration between humans, machines, and algorithms until 2025. Secondly, but related to the previous one, the environmental challenges such as climate change, exponential growth in global populations, and consumption, recognize the need to implement strategies that embrace the green economy vision. Thirdly, the economic recession and labor market issues have increased job insecurity, youth unemployment, and social exclusion. According to the latest data [6], 13% of 15–29-year-olds in the EU in 2021 were neither in employment nor in education or training (NEET). Finally, the COVID-19 pandemic and the related economic crisis led to vast increases in unemployment. As noted by the International Labor Organization [7], youth were the at-most risk category of unemployment already before the COVID-19 era. These global phenomena have induced changes in the labor market bringing new risks and challenges for future workers [8]. Among others, young people from stagnating economies are the first to be affected.

In countries like Italy, the last economic crisis has been dramatic for the careers of young people [6]. Italy has the highest rate of NEET among European countries [6], and represents a unique example in Europe of the gap between the Northern and Southern regions. The Southern workforce is significantly smaller than in the Northern regions, and the school-to-work transition is more difficult and takes much longer [9]. In particular, among the Southern regions, Campania is the region most affected by unemployment [10] and may constitute an interesting context in which to study the way to enhance positive resources to master career transitions. According to Fusco and colleagues [11], in these environments, adolescents and young people might develop harmful psychological outcomes such as skepticism about positive changes in society, cynicism, and hopelessness.

In light of this, the study considers the interplay between career development and positive psychology [12], and outlines the importance of flourishing in adolescence in determining career achievements. Flourishing is the most recent conceptual framework of positive psychology for defining well-being and is conceived as a positive psychological state characterized by positive emotions, engagement, positive relationships, meaning, and accomplishments with various positive work and life outcomes [13]. According to [13], the experiences of fruitful careers can be traced back to the skills and abilities promoted during adolescence and early adulthood, as well as the positive psychological state, or flourishing, experienced during this phase of life.

Given the importance of achieving and maintaining positive levels of flourishing in adolescence, both concerning the individual's present and future well-being, this study aims to investigate some possible predictors of flourishing in the frame of positive career development, i.e., courage, self-esteem, and career adaptability. In this light, this work contributes to the recent strand of studies that consider the possible trade-off between career development and positive psychology frameworks enhancing knowledge about the antecedents of flourishing. Some studies have already shown the role of career resources in predicting flourishing in the young adult population; however, the literature is still scarce for adolescents. For this reason, this study focuses on adolescents who are the most concerned.

### 1.1. The Interface between Positive Psychology and Career Development

Positive psychology may enrich career development and provide a fruitful focus on well-being [12]. The mission of Positive Psychology is the promotion of positive well-being for the whole population by boosting effective functioning. Given the above, according to [14], positive psychology is a social movement with an agenda that overlaps that of career counseling. The purpose, defined as "noble", is the same, i.e., increasing the well-being of individuals and communities. Robertson [12] provided a systematization of the contribution of positive psychology to career development, showing that positive psychology "represents a fertile source of innovative approaches to career counselling, such as the identification of strengths, and the promotion of positive feelings and positive orientations to the future" (p. 256).

To understand the interface between positive psychology and career development, we must consider the central value of work in the identity development of adolescents [15], and the individual's perception of subjective well-being [16–18]. In this view, career choices may contribute to the positive development of the individual with the final goal of psychological well-being. Hence, the integration of career development with positive psychology implies that individuals can be guided during career development in order to attain fulfillment [19].

In a recent article on career development, Wehmeyer and colleagues [20] have claimed that promoting positive career resources, such as career adaptability [21], assumes a preventive aim that contributes to the positive growth of individuals and the social system. Therefore, the interplay of the two theoretical frameworks is necessary to remark on the need for career resources to enhance well-being, i.e., flourishing.

*1.2. Flourishing*

Flourishing arises as an integrated well-being conceptualization [22] encompassing the hedonic and eudaimonic approaches. Hedonic well-being refers to the individual pursuit of pleasure or happiness and involves the assessment of subjective well-being as positive and negative affect balance and life satisfaction [23]; while eudaimonic well-being refers to the actualization of human potentials and involves the assessment of six dimensions of psychological well-being: autonomy, environmental mastery, personal growth, positive relations with others, purpose in life, and self-acceptance [24].

Keyes [25] conceptualized the concept of "flourishing" and "languishing" to describe the highest and the lowest levels of mental health, combining the measures of subjective and psychological well-being. Several authors have tried to identify the underlying dimensions of flourishing [13,22,25,26] until [26] proposed the currently most widely used scale to assess flourishing. Following the theoretical framework of eudaimonic dimensions of the well-being of Ryff [27] and Ryan and Deci [24], Diener and colleagues [26] proposed a concept of flourishing that involves competence, self-acceptance, meaning and relatedness, optimism, giving, and engagement as dimensions of positive functioning.

According to Seligman [13,28], individuals who are flourishing have more chances of having a successful career, are more productive at work, and are more peaceful than people who are not flourishing. Hence, flourishing is the optimal state of functioning in which individuals fulfill their potential and pursue their goals and aspirations.

Despite the growing interest in flourishing (for a review, see [29]), few studies examined the link between career resources and flourishing. Magnano and colleagues [30] underlined the important role of flourishing in the university-to-work transition. Some qualitative studies have shown the importance of flourishing during the transition from high school to university [31,32]. Moreover, other studies have shown the relationship between career adaptability, courage, and life satisfaction in middle and high school students [33,34]. Since, as we pointed out above, in the middle school period, individuals must make the first career choice, it is relevant to study the relationship between career resources and flourishing at this age.

*1.3. Courage*

Traditionally, courage has been investigated as an accolade and a process [35]. The vision of courage as an accolade requires objective, external standards to establish objective risk and define an action as courageous. In this definition, courage is treated as a trait. The vision of courage as a process requires the subjective evaluation of action from the point of view of the individual who experiences it. This definition has been used by several disciplines. For example, Rate [36] defined four primary aspects of courageous behavior: intention, deliberation, risk, and noble (or prosocial) intentions.

Positive psychology has unearthed the construct of courage, debating the nature and dynamics of courage, as well as the possible antecedents and outcomes [37]. Norton and Weiss [38] provided an operational definition of courage as behavior characterized by the persistence of effort despite warning fear.

In this paper, we used this conceptualization of courage following the view that it can be felt like an adaptive behavior for coping with career development tasks and for promoting life satisfaction [33]. Several studies have shown the relationship between courage and different indicators of well-being, such as psychological and subjective well-being in adolescents [39] and adults [40,41]. In a recent paper, Magnano and colleagues [30] highlighted the significant relationship between the measure of courage and flourishing in a sample of college students, pointing out that courage results in behavior that motivate one to face transitional situations, overcome one's fears, and thus contribute to the perception of a meaningful life.

### 1.4. Self-Esteem

Self-esteem is conceptualized as an affective evaluation of (or feeling regarding) the self [42,43]. It is distinct from general self-efficacy, which refers to the individuals' perception of their ability to perform in different situations [43,44]. Self-esteem does not necessarily reflect the objective abilities of individuals but is related to feelings of self-acceptance and self-respect.

Self-esteem has a positive effect on life and career outcomes, such as work conditions [45]. Several studies highlighted that low self-esteem could be a risk for depression in adults [46,47]. Moreover, Trzesniewski and colleagues [48] found that low self-esteem during adolescence predicts poor health and limited economic prospects during adulthood.

Self-esteem is related to career constructs such as career adaptability [34,49], and career expectations [50]. Salmela-Aro and Nurmi [51] examined how self-esteem during university studies impacted the characteristics of the work career ten years later, showing that a high level of self-esteem predicted being in permanent employment, a good salary, and a high level of work engagement and job satisfaction, and a low level of burnout. Instead, a low level of self-esteem predicted being in unemployment, feelings of exhaustion, cynicism, reduced accomplishment at work, and low levels of work engagement and job satisfaction. Finally, several studies showed the relationship between self-esteem and life satisfaction [52,53].

### 1.5. Career Adaptability

Career adaptability is the core dimension of the Life Design paradigm. Life Design is conceived as a lifelong self-construction process that aims to enhance skills and competencies in life planning [54]. Career adaptability refers to a psychosocial resource for managing career-related tasks, transitions, and traumas. According to this approach, with the transition from stability to unpredictability, individuals should increase their adaptability to face unpredictable situations by making changes. The dimensions of career adaptability are (a) concern, (b) control, (c) curiosity, and (d) confidence [55]. The skills in these dimensions include planning, decision-making, exploring, and problem-solving, respectively. Concern refers to the possibility of being positively projected toward the future for creating it desirable. Control refers to the tendency to think that the future is, in part, manageable and the possibility to manage the relationship one will have with the environment. Curiosity refers to a predisposition to explore the environment and the way a career could make it better. Confidence refers to having belief in one's ability to master the challenges that may be encountered in pursuing their goals and the chance of influencing the environment through personal actions.

Promoting career adaptability contributes to the positive growth of individuals. Among several positive career outcomes of career adaptability (for a review, see [56]), studies showed the relationship between career adaptability and optimism, hope, life satisfaction [57], and well-being [58,59].

### 1.6. Current Study

Guided by the interplay between career development and positive psychology, this study analyzed the relationship between courage, self-esteem, and career adaptability, and their role in predicting flourishing in adolescents. Flourishing, as the optimal state of functioning, is conceived as most relevant in determining later career achievements. The study of its predictors could contribute to the growing interest in the scientific literature on promoting flourishing in adolescence. The integration of the two theoretical frameworks, the career construction theory and positive psychology, is called for to emphasize the need for career resources to improve flourishing in adolescence. Previous studies have emphasized the role of flourishing during the transition from high school to university [31,32]. For example, Volstad and colleagues [32] found that fostering personal strengths normalizing challenges experienced in the first year of university, and encouraging students to take on challenges, are critical to supporting their flourishing.

In our study, we focus on career resources, i.e., career adaptability, courage, and self-esteem. While previous studies highlighted a significant relationship between career adaptability, courage, and flourishing among university students [30], there are no studies on the relationship between self-esteem and flourishing. Instead, some studies highlighted an association between self-esteem and career adaptability [34], and between career adaptability and positive outcomes [57–59]. These results suggest that career adaptability might act as a mediator in self-esteem/courage-flourishing relationships.

Moreover, to our knowledge, no studies have explored the link between courage, career resources, and flourishing among adolescents. As described in the previous sections, adolescence is a crucial period in career development. For this reason, it may be important to examine the effects of courage, self-esteem, and career adaptability on flourishing in adolescence. The findings could provide career practitioners with relevant insights for designing career guidance interventions to support career construction in adolescence by taking positive development and thus flourishing as the main goal.

Finally, our study also considers the possible gender difference in flourishing. Literature on gender differences in flourishing is controversial. Some studies reported greater flourishing in males [60,61], while in other studies, these gender differences were present in favor of females [62–67]. Moreover, other studies do not detect a gender difference in flourishing [30,68]. Related to their study of adolescents' sample, Romano and colleagues [69] showed a relatively poor flourishing among females compared to males. Given the widely disparate findings regarding gender differences in flourishing, there is a need for a study in this direction to increase the evidence on the topic.

Considering the above, the main aim of the present study was to analyze the predictive role of courage, self-esteem, and the four career adaptability dimensions (concern, control, curiosity, and confidence) on flourishing. We expected that courage, self-esteem, concern, control, curiosity, and confidence, predict flourishing. Moreover, relationships between courage, self-esteem, the four career adaptability dimensions, and flourishing were explored. In particular, we expected the four career adaptabilities to mediate the self-esteem–flourishing and courage–flourishing relationships.

## 2. Materials and Methods

### 2.1. Participants

A sample of 221 adolescents, 93 males and 128 females (Mage = 13.57, SDage = 0.46), attending the third (last) years of middle school in the Campania Region (Southern Italy), took part in this study. Adolescents filled out the questionnaire booklet in a classroom during their lesson time in the presence of the first author.

Before undertaking the study, informed consent was obtained from the principals of the middle schools and parents. During the data collection, students received information about the aim of the study and were reassured about the confidentiality of their answers. Completion time was between 20 and 30 min. The study was approved by the Ethical Committee of Psychological Research of the University of Naples Federico II.

### 2.2. Measures

Demographic data were obtained from the participants. Students reported their gender (0 = male, 1 = female) and age. In addition, the following self-report measures were administered.

#### 2.2.1. Flourishing

To assess flourishing, we used the Italian version of the Flourishing Scale [26]. The Italian version was validated by Di Fabio [70]. The scale consists of 8 items (e.g., I am optimistic about my future", "I lead a purposeful and meaningful life") rated on a 7-point Likert scale ranging from 1 (strongly disagree) to 7 (strongly agree). Cronbach's alpha of the study sample was 0.91.

### 2.2.2. Courage

To assess courage, we used the Italian version of the Reduced Courage Measure version [38,71]. The Italian version was validated by Ginevra and colleagues [72]. The scale consists of 6 items (e.g., "I tend to face my fears", "If there is an important reason to face something that scares me, I will face it") rated on a 7-point Likert scale ranging from 1 (never) to 7 (always). Cronbach's alpha of the study sample was 0.85.

### 2.2.3. Self-Esteem

To assess self-esteem, we used the Italian version of the Rosenberg Self-Esteem Scale [73]. The Italian version was validated by Prezza and colleagues [74]. The scale consists of 10 items (e.g., "I feel that I have a number of good qualities", "I feel I do not have much to be proud of") rated on a 4-point Likert scale ranging from 1 (strongly disagree) to 4 (strongly agree). Cronbach's alpha of the study sample was 0.87.

### 2.2.4. Career Adaptability

To assess career adaptability, we used the Italian version of the Career Adapt-Abilities Scale [55]. The Italian version was validated by Soresi and colleagues [75]. The scale consists of 24 items rated on a 5-point Likert scale ranging from 1 (not strong) to 5 (strongest). It has 4 dimensions: concern (6 items, e.g., "Becoming aware of the educational and career choices that I must make"), control (6 items, e.g., "Taking responsibility for my actions"), curiosity (6 items, e.g., "Exploring my surroundings"), and confidence (6 items, e.g., "Working up to my ability"). Previous studies have validated the instrument for pre-adolescents (middle school students) in the Italian context, confirming the psychometric properties of the 4-factorial structure [76]. Cronbach's alphas of the study sample for the 4 subscales were 0.89, 0.87, 0.88, and 0.90, respectively.

### 2.3. Data Analysis

Preliminary analyses were performed. First, the internal consistencies of measures were assessed by computing Cronbach's alpha coefficients. Second, means, standard deviations, and Student's *t*-tests were performed to detect differences between males and females in courage, self-esteem, career adaptability, and flourishing levels. The effects of the differences were evaluated with Hedge's *g*. Third, bivariate correlations between courage, self-esteem, career adaptability, and flourishing were computed to have a first insight into associations between the variables. Fourth, to investigate the relationships between courage, self-esteem, career adaptability, and flourishing, SEM path models were set up. In the first model, the four dimensions of career adaptability were set as partial mediators of the relation between, respectively, courage and self-esteem, and flourishing. A second model was tested in which the non-significant relationships were removed. The following fit indices were used to evaluate the fit of the models: CFI and TLI ([77]; good if >0.90), and RMSEA ([78]; good if <0.08).

## 3. Results

Means and standard deviations for the total sample and male and female sub-samples are reported in Table 1. Student's *t*-tests showed statistically significant differences between the male group and the female group in courage (t = −3.225; *df* = 219; *p* < 0.001; *g* = 0.424) and flourishing (t = −3.367; *df* = 219; *p* < 0.001; *g* = 0.723). Females reported higher scores than males on both variables. Scales' reliability is confirmed by Cronbach's alphas ranging between 0.85 and 0.91 (Table 2).

Bivariate correlations among the main variables of the study are reported in Table 2. Correlation analysis showed positive relationships between courage and flourishing and between self-esteem and flourishing. A weaker but significant association also emerged between courage and self-esteem. Correlation analysis also showed a strong relationship between all the career adaptability dimensions. Moreover, the four adaptability dimensions are associated positively with courage, self-esteem, and flourishing.

**Table 1.** Means and standard deviations for the total sample, females, and males.

|  | Total | Males | Females |
|---|---|---|---|
|  | **M (SD)** | **M (SD)** | **M (SD)** |
| Flourishing | 46.185 (7.039) | 43.376 (7.093) | 48.227 (6.277) |
| Courage | 5.529 (0.995) | 5.281 (1.076) | 5.701 (0.893) |
| Self-esteem | 32.115 (3.327) | 31.935 (2.586) | 32.247 (3.780) |
| Career concern | 4.117 (0.695) | 4.088 (0.714) | 4.138 (0.683) |
| Career control | 4.287 (0.576) | 4.297 (0.578) | 4.280(0.577) |
| Career curiosity | 4.270 (0.569) | 4.213(0.544) | 4.312 (0.584) |
| Career confidence | 4.327 (0.574) | 4.244 (0.610) | 4.387 (0.541) |

Note. M = mean; SD = standard deviation.

**Table 2.** Cronbach's alpha and correlations.

|  |  | $\alpha$ | 1 | 2 | 3 | 4 | 5 | 6 | 7 |
|---|---|---|---|---|---|---|---|---|---|
| 1 | Flourishing | 0.91 | - |  |  |  |  |  |  |
| 2 | Courage | 0.85 | 0.56 * | - |  |  |  |  |  |
| 3 | Self-esteem | 0.87 | 0.29 * | 0.18 * | - |  |  |  |  |
| 4 | Career concern | 0.89 | 0.40 * | 0.31 * | 0.24 * | - |  |  |  |
| 5 | Career control | 0.87 | 0.46 * | 0.47 * | 0.27 * | 0.61 * | - |  |  |
| 6 | Career curiosity | 0.88 | 0.46 * | 0.45 * | 0.32 * | 0.64 * | 0.64 * | - |  |
| 7 | Career confidence | 0.90 | 0.52 * | 0.48 * | 0.28 * | 0.60 * | 0.59 * | 0.71 * | - |

Note. $\alpha$ = Cronbach's alpha; * $p < 0.001$.

The first SEM path model tested, including the four career adaptabilities as mediators of the courage–flourishing and the self-esteem–flourishing relationships, was a fully saturated one. In the model, both courage and self-esteem had significant and positive effects on the four career adaptability dimensions and flourishing. However, only the confidence dimension had a significant effect on flourishing. Hence, in the second model (Figure 1), which also showed good fit indices (TLI = 0.98; CFI = 0.99; RMSEA = 0.05), the three links of concern, control, and curiosity with flourishing were removed. Concerning the effect on flourishing of the variables included in the model, courage (0.40) and confidence (0.29) had the greater significant standardized beta values, whereas self-esteem was the lowest (0.13). Overall, courage, self-esteem, and confidence explained 41% of the flourishing variance. Moreover, confidence seemed to partially mediate the effect of courage and self-esteem on flourishing.

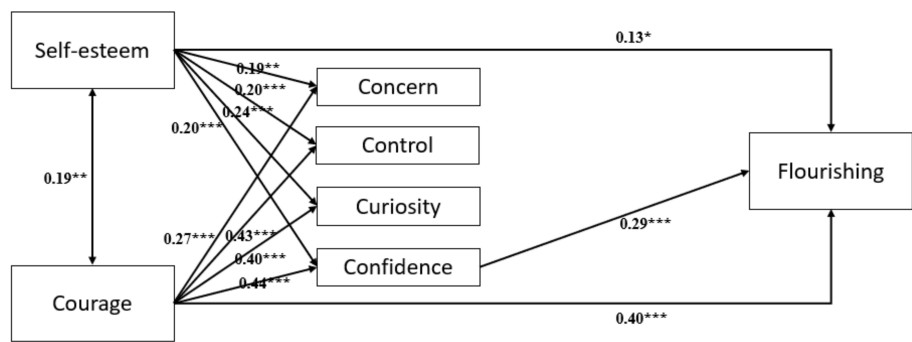

**Figure 1.** Structural equation model. Note. Only significant paths are reported. * $p < 0.05$, ** $p < 0.01$, *** $p < 0.001$.

## 4. Discussion

The in-depth study of personal resources related to flourishing appears timely and significant, especially in the countries most affected by youth unemployment. As mentioned above, flourishing is the optimal state of functioning in which individuals pursue their goals and aspirations. Seligman [13] has pointed out the crucial role of flourishing in achieving successful careers. Despite the growing interest in flourishing, fewer studies explored the link between courage, self-esteem, career adaptability, and flourishing among adolescents. Insights about a potential relationship between these variables come from previous studies made on samples of young adults (for example, [30]). This study aimed to fill this gap by examining the relationship between courage, self-esteem, the four dimensions of career adaptability (concern, control, curiosity, and confidence), and flourishing in adolescents.

Preliminary analyses showed gender differences in courage and flourishing with higher levels of both dimensions in the sample of females. The higher level of courage in females than males is not in line with the previous studies. Ginevra and colleagues [33] showed a higher level of courage in males than females in a sample of high school students, while Magnano and colleagues [30] found no gender differences among university students. Regarding the gender differences in flourishing, existing research is controversial and our results are in line with some studies [65] but not with others [69]. Although there is no widely accepted evidence, Arrosa and Gandelman [79] claimed that females tend to be happier than males worldwide. The authors explained this difference, considering that the female seems to have a more optimistic view of life.

The SEM model revealed that both courage and self-esteem had significant and positive effects on the four career adaptability dimensions and on flourishing and that confidence partially mediates the effect of courage and self-esteem on flourishing.

Regarding the relationship between self-esteem and career adaptability, the findings confirmed the literature [80,81], and suggested that adolescents with a higher feeling of self-esteem also tend to have a higher view of themselves in terms of career adaptability. Moreover, the relationship between courage and career adaptability has been found in previous studies with Italian adolescents [33]. On the relationship between career adaptability and flourishing, although we expected all four dimensions to predict flourishing, results showed that only the confidence dimension has a role in predicting flourishing. Our studies highlighted a substantial difference from the previous study by Magnano and colleagues [30] conducted with university students. Whereas Magnano and colleagues [30] reported a significant effect of career concern, our study showed an effect only on career confidence. This evidence can be explained by taking into account the adolescent stage. Confidence is the extent to which individuals believe in their ability to make good career decisions. This finding reflects that building confidence may be beneficial for the well-being of adolescents and reduce the stress related to career transition. Furthermore, the findings suggested that to "flourish" adolescents should feel they can overcome their fears, accomplish a positive affective evaluation regarding the self, and feel confident about their abilities. These results recommend the importance of cognitive and affective evaluations of the self and their abilities. Consistent with Van Zyl and Stander [18], flourishing interventions should be aimed at developing the needed skills to develop a high level of self-awareness to translate experiences into optimal career-related choices.

Our results suggest the importance of career guidance interventions in the educational systems. These interventions should be oriented toward boosting confidence in adolescents' abilities and achieving their goals. Uncertainty about one's abilities seems to be a risk factor for adolescents. Interventions that help build or strengthen courage, self-esteem, and career confidence among middle school students have the potential to improve mental health in their career transitions. A possible solution for supporting career resources in adolescents is the Life Design intervention. The Life Design approach is a lifelong self-construction process aiming to promote career resources [20,54,82,83]. In a context like Italy, it is important to stimulate career adaptability [84] and all other resources necessary for the

development of creative thinking related to the desire to create the environment in which young people would like to live [11].

This study has also limitations. First, its cross-sectional nature does not permit the inference of reliable causal relationships between the considered variables. Longitudinal studies should be made to put to the test the relations found. Second, all the measures used were self-reported and the data may be influenced by a reporting bias. Third, the sampling unbalanced by gender poses limitations to the generalizability of the results. Related to this, as mentioned above, a less clear-cut picture emerges about the effect that gender may exercise on flourishing. Further studies should investigate in-depth the difference between males and females in flourishing. Fourth, the study refers to a southern Italian sample, so results need to be replicated in other geographical areas to determine their generalizability. The literature suggested that flourishing predictors and outcomes can be unique to a specific population and context [85]. Fifth, the study does not consider external variables that may play a role in flourishing. For example, it might be interesting to study the influence of parents as key figures in adolescents' career development [86]. Finally, future studies might also use a qualitative method for in-depth exploration of the experiences of flourishing [31,32].

## 5. Conclusions

The present study looks at the interplay between career and positive psychology concepts. In our view, positive psychology may enrich the career development approach to cope with the challenges of current society that led to the linear and normative career transitions proving a fruitful "priority" on well-being.

This study investigated the contribution of courage, self-esteem, and career adaptability (concern, control, curiosity, and confidence) in explaining flourishing. Results suggest the crucial role of courage, self-esteem, and career confidence.

These results should be read in the light of the context in which career transitions take place, and the same career interventions for supporting the flourishing of adolescents cannot be conceived in isolation but considering the context.

**Author Contributions:** Conceptualization, A.P. and J.M.; methodology, A.P. and J.M.; formal analysis, A.P. and J.M., writing—original draft preparation, A.P.; writing—review and editing J.M. All authors have read and agreed to the published version of the manuscript.

**Funding:** This research received no external funding.

**Informed Consent Statement:** Informed consent was obtained from the principals of the middle schools and parents.

**Data Availability Statement:** The data presented in this study are available on request from the corresponding author. The data are not publicly available due to privacy issues.

**Conflicts of Interest:** The authors declare no conflict of interest.

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
