# Peer review of "Positive Resources for Flourishing: The Effect of Courage, Self-Esteem, and Career Adaptability in Adolescence"

_societies, doi:10.3390/soc13010005_

Round 1

Author Response

I want to thank the editors for allowing me to review this paper. The paper presents a study testing the relationships between gender, career resources (self-esteem, perceived courage and career adaptability) and flourishing. Despite not being groundbreaking, this paper has the merit of progressing knowledge about the antecedents of flourishing. I appreciate that the study is timely with youth's global and local employment scenario and the need to enhance psychological well- being about (future) careers. I appreciate the focus on young people who are the most concerned, especially in these pandemic times, and the attempt to integrate career notions with suggestions from the positive psychology framework, which may also be translated into practical suggestions. Despite these positive features, I have some severe concerns at all levels (theoretical, methodological and in terms of contribution) that, at the moment, I consider highly necessary to address to improve the quality of this work and make it publishable. I hope these comments can be helpful. To make it easier for authors to move through the comments, I have distributed them using the paper sections.

We would like to thank you for your feedback. We have tried to address the concerns raised and we hope you share our impression that the quality of the manuscript has improved substantially. Following, the response point by point. All changes/additions are highlighted in yellow in the manuscript

  1. Introduction

1)  I appreciate the description of the youth employment scenario. However, I’ve noticed that in the discussion section (see p. 7, lines 283 – 292), there’s a focus on the employment situation in southern Italy. It would be nice to move this focus up in the introduction to frame the societal problem better. Also, and this is of the utmost importance, authors should explain why it is necessary to focus on a sample of adolescents. This is required to enhance the solidity of this study’s rationale, which is quite weak here. 

Thank you for both comments. Firstly, we moved the focus on the employment situation in southern Italy in the introduction. Secondly, we have explained why it is necessary to focus on a sample of adolescents in the first sentences of the manuscript.

2)  In my opinion, the introduction should (at least briefly) mention the theoretical framework used (the interface between career development and positive psychology framework), so the reader can have anticipation. 

Thank you for this suggestion. We have anticipated the theoretical framework used in the introduction section.

3)  I’d leave aside the reference to counselling and intervention to remark on the importance of flourishing (p. 1, lines 43-44). It may be enough to outline flourishing importance in determining later career achievements.

We modified the sentence following this suggestion. Thank you.

4)  I also suggest the authors anticipate the novelty/contribution of the study, both at a theoretical/research level and a practical one.

Thank you for this suggestion. We have anticipated the novelty of the study in the introduction section.

5)  I’d clarify better, in paragraph 1.1, that the integration of the two theoretical frameworks are necessary to remark on the necessity to have career resources to enhance well-being (i.e. flourishing) 

We agree with you in this statement, which captures our point of view. We have added this sentence as a conclusion to section 1.1

6)  Could paragraph 1.2 be reduced in text by defining flourishing and briefly mentioning the distinction between Hedonic and Eudaimonic well-being approaches? Moreover, I think that some of the evidence reported in the discussion section about flourishing (p.7, lines 293- 306) could be used here to report and analyse current evidence on flourishing and explain what gaps you intend to address. 

Thank you for this suggestion. We have reduced the part on hedonic and eudaimonic approaches focusing on flourishing. Moreover, as indicated, we have moved the studies in the discussion to this section to highlight the gap in the literature.

7)  The role of courage in fostering flourishing should be better explained in paragraphs 1.3 or 1.6. 

Thank you for this suggestion. We have specified the role of courage in fostering flourishing in the paragraph 1.3.

8)  Why it is important to remark on the stability of self-esteem (p. 3, lines 127-131)?

Thank you for this suggestion. We have removed this aspect because out of scope of this article.

9)  Paragraph 1.6 should be more compelling on the reason why conducting this study. What is 
the problem here? Why should exploring career resources/flourishing relationships be studied among adolescents when there’s already consistent literature supporting this kind of relationship, even with different samples? It is only stated that “To our knowledge, no studies have explored the link between career resources and flourishing among adolescents”. Why should this be relevant?

Thank you for these remarks. We have modified the whole section following the comments.

10)  Page 4, lines 171-173, the relationships studied should be stated more clearly. 

Thank you for this comment. We have added more information about the studies.

11)  Page 7, line 178. The role of gender is introduced. While it could be noteworthy to explore 
gender differences, it should at least be explained why. Why is it important to explore gender differences? What does it add? Moreover, if gender is used here, it should be included in the hypothesis. Also, the discussion section (p. 8, lines 325-335) presents a brief literature review on gender and flourishing, which may belong more here than to the discussion section.

Thank you for this comment. We have moved the literature from the discussions to this section and clarified why we included it.

12)  I’m not very convinced about H1. Is it really necessary to formulate and test such a hypothesis? Perhaps most career studies literature has reported positive relationships between career resources. This is compellingly supported also at a theoretical level. For instance, the notion of resource caravans from the Conservation of Resources theory assumes that psycho-social resources do not exist singly but move in pools (or caravans) and for such reason, resources are positively correlated. That said, I really cannot see why it can be useful to have such a hypothesis.

Thank you for these comments. We have delated the H1.

13)  Page 4, lines 190-192. This seems quite out of position here. I'd suggest moving it before at the end of the introduction section.

Done! We moved the sentence at the end of the introduction section.

  1. Method

14)  Page 4, lines 199-200. Is this inclusive of informed consent? Informed consent procedure should be clearly stated.

Yes, we modified the sentence to make it clearer.

15)  Please report examples of the items of each scale used. 

We have added examples of items for all measures used

16)  Haven’t the authors collected any control variables (e.g. socioeconomic background)? How was gender coded in this study?

We requested only gender and age. The school did not allow us to collect other data, such as socioeconomic background because it was considered "sensitive data." We reported how we coded gender in the measures section.

17)  Page 5, lines 233-235. This is not anticipated by the hypotheses/research question.

According also with the second reviewer we have performed the Students’ t-tests instead of ANOVA. Moreover, t-tests and correlations are added as preliminary analyses. Thus, no hypothesis (in line also with the suggestion related to the correlation) were formulated.

18) and 20) I'm not sure why performing two hierachical regressions. This should be stated clearly. Moreover, why courage and self-esteem were included together?

20)  I have a very serious concern about the H2 results report. This report needs to be clarified. It seems to mix results from the two hierarchical regressions. For clarity, please report the results for both hierarchical regressions separately, which also apply to tables. Moreover, I need clarification on why such hierarchical regression results should suggest an indirect effect. I'm not sure you can infer this kind of conclusion based on the results you obtained. At least you should perform a mediating analysis. Otherwise, you can use this as a suggestion for additional research in the discussion section. I think that this results’ part should be rewritten to enhance clarity and coherence with the results that you obtained. It severely affects the readability of the manuscript. 

Thank you. In light of these comments, we decided to perform a SEM path model permitting to model and test more clearly the hypothesized relationships. Therefore, we modified the data analysis and results section.

19)  In reporting results, authors should state whether or not they support their hypotheses.

Thank you, we discussed the results considering the hypotheses.

Hirschi, A. (2012). The career resources model: an integrative framework for career counsellors. Br. J. Guid. Couns. 40, 369–383. doi: 10.1080/03069885.2012.700506

Thank you for this reference. We have added this work in the manuscript.

  1. Discussion

21) The first part of this section, from lines 283 to 306, could be sensibly reduced in words. Most of this part reads like a literature review rather than a critical analysis of the results. 491- 499. This may belong better to the formulation of the hypothesis.

We have completely rewritten the discussion part (also in accordance with all the suggestions of moving the literature from the discussion to the introduction sections).

22)  Page 7, lines 311-314. does this confirm hypothesis 1 or not? it should be clearly stated.

We eliminated hypothesis 1 as suggested.

23)  Page 8, lines 315-335. Again, I'd expect to see this in the background. You can briefly mention previous literature here (in terms of alignment or not with your results), yet here I'd like to see a critical explanation of your results.

We have completely rewritten the discussion part (taking in to account also this suggestion)

24)  Page 8, lines 346. I need help seeing the connection between confidence and self-esteem here. If confidence enhances flourishing, why in virtue of this self-esteem, should do the same?  

We have clarified this aspect in the discussion section.

25)  Page 8, lines 352-355. I still need to see how you can infer such an indirect effect without testing it. 

As clarified in the comments above, we performed a SEM path model.

26)  Can you provide one or two examples of interventions aimed at Italian adolescents to enhance career resources?

Thank you for this suggestion. We did that in the final part of the discussion.

  1. Additional suggestions

Please, make an extensive and conscious revision of the manuscript's language, grammar and style to make it more readable. Some typos and repetitions are present throughout the paper.

Thanks for the suggestion. The paper was reviewed by a native English

Reviewer 2 Report

*This is not my field of study, methodologically at least, so I was pleased to read that the author(s) acknowledge some of the limitations of the reported study, not least the ‘external variables’ that may well contribute to the characteristics reported by the sampled respondents.  Nonetheless, I found the paper very interesting, despite continuing to have reservations about the concept of ‘self-esteem’ (following Nick Emler’s now quite dated review of the literature on it).*

**

*There are some very minor presentational issues: a clumsy sentence on line 23 and the repeated use of ‘Moreover’ in lines 175 and 178.  Otherwise, the paper is very clearly structured and written.*

**

*My only substantive concern/criticism is the discussion of gender, its absence from the title of the paper, and the positioning of that discussion in the paper.  Given its prominence within the paper, gender should be in the title.  Furthermore, I think that much of the discussion about the contested nature of the effect of gender on other variables discussed (lines 315-335) should be presented earlier in the paper (around 1.2), by way of introduction.  Its current position muddles the discussion/analysis, in my view.*

**

*I was not aware of the concept of ‘flourishing’ (or ‘languishing’) before now.  The discussion within the paper is very reminiscent of earlier debates about ‘resilience’ and ‘risk’; this paper sustains that debate in relation to youth transitions, careers guidance and labour market prospects.* 

Author Response

This is not my field of study, methodologically at least, so I was pleased to read that the author(s) acknowledge some of the limitations of the reported study, not least the ‘external variables’ that may well contribute to the characteristics reported by the sampled respondents.  Nonetheless, I found the paper very interesting, despite continuing to have reservations about the concept of ‘self-esteem’ (following Nick Emler’s now quite dated review of the literature on it).

Thank you for these positive evaluations, which are very flattering to us! We are glad you liked our work. Moreover, we would like to thank you for your suggestions. We have tried to address the concerns raised and we hope you share our impression that the quality of the manuscript has improved substantially. Following, the response point by point.

1) There are some very minor presentational issues: a clumsy sentence on line 23 and the repeated use of ‘Moreover’ in lines 175 and 178.  Otherwise, the paper is very clearly structured and written.

Thank you. We revised the manuscript in light of these suggestions.

2) My only substantive concern/criticism is the discussion of gender, its absence from the title of the paper, and the positioning of that discussion in the paper.  Given its prominence within the paper, gender should be in the title.  Furthermore, I think that much of the discussion about the contested nature of the effect of gender on other variables discussed (lines 315-335) should be presented earlier in the paper (around 1.2), by way of introduction. Its current position muddles the discussion/analysis, in my view.

Thank you for this suggestion. We have moved the literature on gender differences to the introduction.

3) I was not aware of the concept of ‘flourishing’ (or ‘languishing’) before now.  The discussion within the paper is very reminiscent of earlier debates about ‘resilience’ and ‘risk’; this paper sustains that debate in relation to youth transitions, careers guidance and labour market prospects.

Thank you again!

Reviewer 3 Report

Thank you very much for the opportunity to review the manuscript entitled „Positive resources for flourishing: the effect of courage, self-esteem and career adaptabilities in adolescence “. The article is of interest for the readership of Societies as it evaluates the influence of various predictors for flourishing in adolescence.

I would like to encourage authors to consider several issues to be improved. I believe that after incorporating these issues, the paper might have a value for this journal. I hope that my comments are useful for authors, as they further develop the manuscript.

First, I am concerned regarding the use of the described measures for the population of adolescents, especially in the case of Career adaptability. I suggest the authors to refer to this issue when they include information on the validation of the measures by previous studies.  

Second, Anova analysis is usually applied for three or more categorical, independent groups. In the case of two groups (e.g. females and males), independent-samples t-test is more commonly used. Also, I suggest to the authors to clarify in the Table 3 how gender has been introduced in the regressions (probably, the female category).  

Third, in my opinion, the conclusion stated at the end of the Results section has no empirical support: “The hierarchical regressions performed suggest that confidence might partially mediate the relationship between courage and self-esteem and flourishing”. Such a conclusion would require a mediation analysis which is not the case here.

I hope that my comments are useful for the authors in order to revise their study.

Author Response

Thank you very much for the opportunity to review the manuscript entitled „Positive resources for flourishing: the effect of courage, self-esteem and career adaptabilities in adolescence “. The article is of interest for the readership of Societies as it evaluates the influence of various predictors for flourishing in adolescence.

I would like to encourage authors to consider several issues to be improved. I believe that after incorporating these issues, the paper might have a value for this journal. I hope that my comments are useful for authors, as they further develop the manuscript.

We would like to thank you for your feedback. We have tried to the concerns raised and we hope you share our impression that the quality of the manuscript has improved substantially. Following, the response point by point.

First, I am concerned regarding the use of the described measures for the population of adolescents, especially in the case of Career adaptability. I suggest the authors to refer to this issue when they include information on the validation of the measures by previous studies.

Thank you for this suggestion. In the Italian context, the career adaptability measure works very well. We are informed about the current debate on career adaptability in other contexts. As suggested, we mentioned the CAAS psychometric validation study for adolescents (middle school) in the description of the measures. Thank you!

Second, Anova analysis is usually applied for three or more categorical, independent groups. In the case of two groups (e.g. females and males), independent-samples t-test is more commonly used. Also, I suggest to the authors to clarify in the Table 3 how gender has been introduced in the regressions (probably, the female category).

Thank you for the suggestions. Student’s ttests were performed instead ANOVAs. The effects of the differences were evaluated with Hedge’s g. In addition, we included how the gender variable was coded.

Third, in my opinion, the conclusion stated at the end of the Results section has no empirical support: “The hierarchical regressions performed suggest that confidence might partially mediate the relationship between courage and self-esteem and flourishing”. Such a conclusion would require a mediation analysis which is not the case here.

Thank you for this comment. In fact, our sentence was too strong. Therefore, also in agreement with reviewer 1, we modified the analyses and conducted an SEM model estimating the mediation effect.

I hope that my comments are useful for the authors in order to revise their study.

Yes! Thank you again!

Round 2

Reviewer 1 Report

I would like to thank the authors for their conscious revising work. I've found that the amendments they have performed meet the concerns that I have raised. In my opinion, the study now is way sounder at addressing societal problems, connecting and advancing the frameworks adopted and reaching results with a more compelling analytical method and strategy.

That said, and since I have no other issue to disclose, I recommend this study for publication.